# Inadvertent Detachment of Stent Retrievers during Mechanical Thrombectomy—A Clinical and Biomechanical Perspective

**DOI:** 10.3390/life11070658

**Published:** 2021-07-06

**Authors:** Piotr Piasecki, Marek Wierzbicki, Piotr Tulik, Katarzyna Potocka, Adam Stępień, Jacek Staszewski, Aleksander Dębiec, Jerzy Narloch

**Affiliations:** 1Department of Interventional Radiology, Military Institute of Medicine, 04-141 Warsaw, Poland; mwierzbicki@wim.mil.pl (M.W.); jnarloch@wim.mil.pl (J.N.); 2Faculty of Mechatronics, Institute of Metrology and Biomedical Engineering, Warsaw University of Technology, 02-525 Warsaw, Poland; piotr.tulik@pw.edu.pl (P.T.); katarzyna.potocka.stud@pw.edu.pl (K.P.); 3Department of Neurology, Military Institute of Medicine, 04-141 Warsaw, Poland; astepien@wim.mil.pl (A.S.); jstaszewski@wim.mil.pl (J.S.); adebiec@wim.mil.pl (A.D.)

**Keywords:** stent-retriever, pRESET, inadvertent detachment, mechanical thrombectomy, Tigertriever

## Abstract

Background: The inadvertent detachment of stent retrievers during mechanical thrombectomy is an extremely rare but feared complication associated with poor clinical outcomes. We discuss management considerations after an unexpected disconnection of the pRESET stent retriever during mechanical thrombectomy, based on clinical experience and mechanical and phantom studies. Methods: We present a clinical course of rare accidents of stent-retriever separation inside an intracranial vessel that occurred in patients in a comprehensive stroke centre between 2018 and 2020. We designed a phantom study to assess the Tigertriever’s ability to remove a detached stent retriever from intercranial vessels. In the mechanical study, several types of stent retrievers were evaluated in order to find the weakest point at which detachment occurred. Results: Two patients (~0.7%) with inadvertent stent-retriever detachment were found in our database. Failed attempts of endovascular removal with no recanalization at the end of procedure were reported in both cases. mRS after 3 months was three and four respectively. In the mechanical study, the Tigertriever was the most resistant to detachment and was followed by Embotrap > pRESET > 3D Separator. In the phantom study, the pRESET device detached in a configuration resembling the M1 segment was successfully removed with the Tigertriever. Conclusions: Conservative management of the inadvertent detachment of stent retrievers during mechanical thrombectomy in large vessel occlusion may be acceptable in order to avoid further periprocedural complications after unsuccessful device removal attempts. Based on the phantom and mechanical studies, the Tigertriever may be a useful tool for the removal of detached pRESET devices.

## 1. Introduction

Mechanical thrombectomy (MT) has proven to be an effective and safe method of treating large vessel occlusion [1,2,3,4,5]. Device-related complications of MT are estimated to be at around 10% [6,7,8,9,10]. Inadvertent detachments of stent retrievers (SR) are reported to occur at a rate of less than 1%, yet are associated with poor clinical outcomes, consequent to a large stroke area related to unsuccessful thrombectomy. Poor clinical sequelae could be related to detached SR removal attempts (i.e., subarachnoid haemorrhage) [6,11,12].

Inadvertent SR detachments were mostly observed when an older generation of stent retrievers were used for stroke treatment. Due to widespread use in clinical trials, an electrolytically detachable Solitaire AB (ev3 Inc., Irvine, CA, USA) was involved in most reports, with only single instances describing other devices (i.e., pRESET) [6,7,8]. With the newer generations of non-detachable Solitaire devices, there has been a significant decrease in the reports of such complications [11].

There are two treatment options in the case of SR disconnection in the cerebral artery during MT: (a) to remove a detached SR or (b) to treat conservatively. The first option is advantageous for the patient but linked with a high risk of further adverse events, such as vasospasm, artery damage and severe bleeding. There is no dedicated device designed for SR removal from the cerebral artery and, to date, there have been single reports of successfully using the Alligator Retrieval Device (Chestnut Medical Technologies, Menlo Park, CA, USA), snares, another stent or an SR in such cases [13,14,15].

To the best of our knowledge, there are no published management guidelines regarding inadvertent detachment of a pRESET or other SRs, nor has this issue been previously investigated from a biomechanical perspective. We present the results of conservative treatment in patients after inadvertent pRESET detachment during mechanical thrombectomy in our institution. We designed an experimental study using a vascular phantom to assess the feasibility and effectiveness of the Tigertriever SR for the removal of a detached pRESET device from the cerebral middle artery. The Tigertriever and other SRs used in our department were investigated in a laboratory tension test in order to find the potential location of their rupture.

## 2. Methods

### 2.1. Patient Analysis

#### Mechanical Thrombectomy

Patients with acute ischemic stroke were qualified for endovascular treatment according to the 2018 AHA/ASA and ESO-ESMINT Guidelines [16,17]. The type of anaesthesia utilised during the procedure depended on the discretion of the anaesthesiologist and patient’s Glasgow Coma Scale (GCS) status. Patients’ neurological dysfunction and functional status were assessed with the NIH Stroke Scale (NIHSS) and the modified Rankin Scale (mRS), respectively. Functional status was prospectively followed for 3 months. All patients received standard secondary stroke prophylactics and diagnostic procedures according to the guidelines. The STROBE case-control reporting guidelines were used [18].

### 2.2. Laboratory Traction Load Test

#### Study Design

The aim of the tension load test was to find the weakest point (point of failure) and compare stent retrievers with different structure designs. The following types of stent retrievers were tested: Tigertriever (RapidMedical, Yoqneam, Israel); pRESET (Phenox, Bochum, Germany); 3D Separator (Penumbra, CA, USA); and Embotrap (Neuravi/Johnson&Johnson, Galway, Ireland).

The stent retrievers were deployed and fixed in a straight silicone tube with a 3.0–4.0 mm interior diameter and attached to the diagnostic radiography table using a U-holder. The U-holder was clenched in the third proximal part of the SR (~10 mm from the connection of the stent retriever to the pusher wire). Weights were sequentially attached to the loop made on the stent’s wire. Weights were hung for 60 s in the following planned order: 227 g; 337 g; 447 g; 557 g; 668 g; 778 g; 831 g; 941 g; 1051 g; 1144 g; 1254 g; and 1364 g. All of the measurements were conducted using a fluoroscopy system equipped with a vision system (Flexavision HB, Shimadzu). The load leading to stent rupture and stent rupture point were recorded. The maximum force (F) leading to SR rupture exerted by the suspended static load was measured in newtons (N).

### 2.3. Phantom Study

#### Study Design

The aim of the study was to assess the utility of the Tigertriever (Rapid Medical, Yoqneam, Israel) in the removal of a detached pRESET (Phenox, Bochum, Germany) device from the vessel. A 3D-printed vascular model made by stereolithography with a translucent material reproducing the middle cerebral artery (MCA), anterior cerebral artery (ACA) and internal carotid artery (ICA) was used. The diameter of the MCA was 3.0 mm and the diameter of the ICA varied from 3.5 to 5.0 mm (from distal to proximal) (Figure 1D). The phantom was continuously flushed by injecting water heated to 37 °C to simulate the in vivo expansion temperature for the nitinol devices. Phantom studies were performed under fluoroscopy using a monoplane angiography system (GE Innova 4100, Boston, Massachusetts). The 4 × 20mm pRESET (Phenox GmbH, Bochum, Germany) was detached from the pusher wire and placed in the phantom vessel representing the position of the M1 segment of the middle cerebral artery (Figure 1A). The mechanical thrombectomy system, with the guiding catheter Neuron Max (Penumbra, CA, USA), intermediate catheter ACE 64 (Penumbra, CA, USA) and microcatheter Headway 21’ (Microvention, CA, USA) with microwire Traxcess (Microvention, CA, USA) was introduced inside phantom vessels.

The microguidewire with a microcathether passed through the detached pRESET. Then, the Tigertriever was deployed within the pRESET (covering the entire length of the pRESET) (Figure 1B,C).

The pRESET removal procedure was repeated five times by each of the three fully trained neurointerventionalists experienced in MT. The times of the procedures, effectiveness (measured as successful removal) and possible complications (e.g., ×. pRESET migration) were evaluated.

### 2.4. pRESET

According to the manufacturer’s information, the pRESET (Phenox, Bochum, Germany) is a nitinol stent, non-detachable, with radiopaque markers—one in the proximal part and two in the distal part, attached to a 180 cm pusher wire. The device has a closed ring in the proximal segment and dual-type SR cells design for stabilizing the structure of the stent, which is especially useful in tortuous vessels [19]. The pRESET is produced in three sizes, those being 4–20 mm, 5–40 mm and 6–30 mm which work with 0.021” microcatheters—dedicated for MT in the case of carotid-T occlusion and occlusion of the proximal segments of the middle cerebral artery (MCA). Kurre W. et al. estimated the risk of periprocedural complications associated with unexpected detachment of the pRESET stent at 0.7% [6].

### 2.5. Tigertreiver

The Tigertriever (Rapid Medical, Yoqneam, Israel) used in this study is a new stent retriever with a unique construction: collapsible, non-detachable and fully retrievable, with the proximal end of the core wire connected to a slider in the handle. The stent construction is expanded by pulling a core wire, which is connected to the distal end of the mesh. The operator can expand and contract the mesh to conform properly to the diameter of the affected vessel wall [20]. There are no data regarding detachment of the Tigertriever during MT.

## 3. Results

### 3.1. Case Study

We found two cases (<0.7%) of detached pRESET out of 298 analysed mechanical thrombectomy procedures between 2018 and 2020 that occurred to two different interventionalists (20 and 5 years of experience, respectively).

#### 3.1.1. Patient No. 1

An 82-year-old female patient underwent MT under sedation due to sudden onset of left hemiplegia with global aphasia; NIHSS on admission was 15 points. Symptoms of stroke had occurred nearly 120 min before the patient arrived at the Emergency Department (ED). The patient had a history of hypertension, atherosclerosis and paroxysmal atrial fibrillation with no anticoagulation therapy. The initial non-enhanced head CT scan revealed a hyperdense right middle cerebral artery in segment M1, with ASPECTS of 6. CT angiography revealed diffuse intimal atherosclerotic lesions with mixed plaque character (soft–hard plaque) involving the aorta, cephalic arteries of the aortic arch and intracranial arteries, and confirmed the occlusion of the right middle cerebral artery at the M1 segment with low partial collateralization of the ischemic site (grade 1 on the ASITN/SIR Collateral Score) [21]. Intravenous recombinant tissue plasminogen activator (IV rtPA) was administered with a dose of 0.9 mg/kg of patient body weight (10% in the loading dose and 90% in continuous infusion). Endovascular MT using a pRESET 4 × 20 mm stent retriever was initiated after 145 min from the onset. The M1 segment of the right middle cerebral artery was still occluded after the first pass (mTICI 0). The procedure was repeated with the same device. During the second SR retraction, an unexpected pRESET detachment from the pusher wire occurred, preceded by an increase in stent-retriever traction resistance and struts extension. Both the traction and elongation did not exceed the degree typical of the device observed during previous MT. Unsuccessful attempts to retract the disconnected pRESET using another SR were performed (Aperio 3.5 × 28 mm, Acandis Gmbh, Pforzheim, Germany). The Aperio SR could not be deployed successfully, and since the lumen of the pRESET would not pass the microcatheter at the end of the procedure the mTICI was 0. After the procedure, the patient showed no neurological deterioration and NIHSS was assessed at 15 points. At 24 h, a postprocedural non-enhanced head CT scan was performed, which revealed the stent retriever placed in the distal C7 segment of the ICA and its ends on the border of segments M1/M2 of the MCA, with ASPECTS of 5 (Figure 2A,B). Finally, single antiplatelet therapy (ASA 150 mg o.d.) was administered after exclusion of haemorrhagic stroke transformation. After 7 days, the patient was assessed and scored 7 points on the NIHSS. The modified Rankin Scale (mRS) score after 90 days was 3.

#### 3.1.2. Patient No. 2

An 88-year-old female patient was admitted to the ED with left hemiparesis, aphasia and a loss of consciousness—NIHSS on admission was 20. Concomitant diseases of the patient were hypertension, atherosclerosis and paroxysmal atrial fibrillation with anticoagulation therapy with acenocumarol. Intravenous thrombolysis was not administered due to the therapeutic level of anticoagulation. An emergency non-enhanced head CT scan and CT angiography demonstrated occlusion of the left MCA M1 segment, with no established infarction in its supply (ASPECT 10). Moreover, CT angiography revealed diffuse intimal atherosclerotic lesions with mixed plaque character (soft–hard plaque) involving the aorta, carotid arteries, vertebral arteries and intracranial arteries. The exact time of onset to groin puncture was 182 min. Angiographic examination under general anaesthesia confirmed the findings of CT angiography. Mechanical thrombectomy was performed using a Tigertriever 17 (Rapid Medical, Yoqneam, Israel) as a first-line device with no recanalization effect (mTICI 0). After the first pass, the Tigertriever mesh was partially damaged—the mesh was distorted and the core wire was rendered non-functional. In the next stage, another stent retriever (pRESET 4 × 20 mm) was chosen and deployed at the occlusion—embracing the thrombus in the middle segment of the struts. The first phase of retraction was accompanied by considerable tension, but without complete retention of the SR. After approximately 10 mm of withdrawal, the SR unexpectedly detached from the pusher wire (no excessive elongation was observed). The procedure was ceased after a number of unsuccessful attempts to cross through the detached SR with a microcatheter. The final angiography showed no recanalization with mTICI 0. At 24 h, a postprocedural follow-up CT and CT angiography scan was performed and revealed the stent retriever placed in the middle part of MCA, ASPECTS 3 and no haemorrhagic stroke transformation. Single antiplatelet therapy (ASA 150 mg o.d.) was administered. After 7 days, the patient was assessed at 13 points on the NIHSS. The patient was discharged to the neurological rehabilitation department for further treatment and the modified Rankin Scale (mRS) score after 90 days was 4.

### 3.2. Laboratory Traction Load Test

The SRs most resistant to inadvertent detachment in the laboratory tension test were: Tigertriever > Embotrap > pRESET > 3D Separator. (Table 1). The Tigertriever did not rupture or detach from the pusher wire; only partial damage to its mesh structure was observed. Some mesh struts were elongated or broken; it was not possible to operate the mesh properly. The Embotrap stent rupture occurred at relatively high loads, which led to rupture of the proximal or distal segments of the stent structure. The pRESET stent rupture occurred at the proximal marker site and in the proximal third segment of the stent. The 3D Separator fractures occurred near its proximal marker (Figure 3A–D).

### 3.3. Phantom Study

All attempts (15/15) resulted in successful removal of the detached pRESET stent retriever (Mov.1). The average procedure times for the removal of the detached pRESET (rescue mechanical thrombectomy procedure) were 165, 173, and 183 s, respectively. Duration and effectiveness of SR retrieval using the Tigertriever were comparable regardless of the experience of the operator. We observed slight tightening of the phantom vessel system during retrieval of the Tigertriever anchored inside the detached pRESET stent. No other disturbing observations were noted. Each attempt of the single SR rescue procedure proceeded without complications (i.e., separation between Tigertriever and pRESET, pRESET migration or damage to the Tigertriever) (Table 1).

## 4. Discussion

Uncontrolled detachment of SRs is associated with many conditions. Atherosclerosis leading to stiffening and stenosis of the intracranial vessel walls, and changing their curvature is one of the major factors increasing the risk of stent retriever disconnection [22]. Hard calcified embolic material/clots, longer time from onset to procedure and inappropriate SR size may be other causes of this adverse event [22,23,24]. The mentioned risk factors were partially found in our patients. Inadvertent pRESET detachments took place in older patients (over 80 years of age) with difficult and challenging cerebral arterial anatomies. Both patients had paroxysmal atrial fibrillation shortly before stroke, which would suggest harder cardiac emboli as a cause. Mechanical thrombectomy was performed less than three hours from onset, using stent retrievers with a size dedicated to proximal vessel occlusions (M1 segment of middle cerebral artery). The pRESET detachments were not anticipated by two different experienced interventional radiologists performing MT in these cases. There were no disturbing signs such as excessive elongation of the stent retriever or excessive cerebral arteries tension, nothing that is not routinely seen with this kind of SR. However, elongation and tension of SR should be carefully assessed during MT as signs of imminent stent disconnection. Since there is no point of reference for what could be regarded as an acceptable range of elongation or tension, interventionalists rely solely on their experience. In order to identify and investigate early signs of possible SR detachment, we performed laboratory tension load tests with the SRs used in our department. We ventured to locate the weakest points of stent retrievers, in which disconnection may occur. In the given loading setup, it was observed that each SR has a unique site and load leading to failure. For the pRESET, in three out of five cases, this was the point of its connection to the pusher wire (as in our clinical situation) and in two cases in its 1/3 proximal part. For the 3D Separator it was in the location of its proximal marker 5/5 times. Interestingly, even though the Tigertriever withstood the highest load, it did not result in its rupture, but led only to SR mesh elongation and partial destruction (Figure 3A). Repeated MT with the same device might lead to its fatigue [12]. Following expert recommendations, we used a new SR after four failed passes [25]. However, this careful approach did not prevent pRESET detachment, which occurred during the second pRESET pass in the first patient and during the first pass in the second patient. In both cases, there were unsuccessful attempts to remove the detached pRESET device by means of other SRs. Finally, conservative management was applied and the middle cerebral arteries were left without restored blood flow. There were no other adverse events related to rescue maneuvers. After exclusion of hemorrhagic transformation in control CT (post 24 h), 150 mg of aspirin was administered. Patient no. 1 was discharged from hospital with an NIHSS of 7 and patient no. 2 with an NIHSS of 13. Their mRS after 3 months was three and four, respectively. Considering the increasing number of MTs, we believe that each interventional radiologist performing such procedures should be familiar with interventional or conservative management of inadvertent stent-retriever detachment. The optimal option is to remove the detached device from the cerebral artery and to continue mechanical thrombectomy in order to restore blood flow [13]. This could be achieved using either other types of SRs to trap the detached stent (‘stent-based retrieval’ technique) [14]; by using alternative devices (e.g., the Alligator retrieval device, Merci retrieval system or endovascular snares); or finally with techniques such as stent retrieval with additional fixation or pull-back manoeuvres with a balloon. In some cases, the use of the ‘deploy and engage’ and ‘loop and snare’ techniques described by Parthasarathy et al. may be the best option [15], especially when the proximal edge of the detached stent is accessible/released, and the stent cannot be crossed with a microwire to attempt removal with another device. There is no dedicated equipment on the market designed to remove a detached or broken SR. Therefore, we assessed the Tigertriever as a rescue device in such circumstances. The main advantage of this stent retriever is its construction, allowing an operator-dependent change of the Tigertriever mesh size to conform properly to the diameter of the affected vessel wall. This feature seemed useful in detached stent-retriever removal. Additionally, we proved in the laboratory tension test that the risk of unexpected Tigertriever detachment is minimal. In the phantom study, we successfully removed the pRESET stent by means of the Tigertriever. This technique could be used for other ‘stent-like’ stent retrievers available on the market. Its main limitation is the need to pass through a disconnected SR with a microcatheter in order to properly position the Tigertriever. Repeated unsuccessful rescue attempts could lead to perforation or dissection of the vessel wall. In the case of good collateral circulation to the brain ischemic area in angiography and low-moderate clinical signs of stroke, we can also decide to leave the stent in the patient’s vessel. Otherwise, even open neurosurgery may be considered as a bail-out option. [26]. A detached SR with integrated thrombus does not pose an additional risk of increased thrombogenicity when the target vessel remained closed; therefore, there is no need to administer antiplatelet therapy. The use of aggressive antiplatelet therapy in the early hours of large-vessel-occlusion-related stroke, especially after administration of intravenous thrombolysis, may lead to an intracranial haemorrhage [27]. In the case of protrusion of the SR fragment into the lumen of the vessel, it is advisable to include dual antiplatelet therapy after prior exclusion of haemorrhagic transformation of the stroke. In this case, in our centre, we use the above-mentioned drugs acc. regimen: ASA 1 × 75 mg and clopidogrel 1 × 75 mg, to prevent stroke in the new territory.

There are limitations of our study. We did not manage to retrieve the detached pRESET in a clinical scenario; the cases were analysed retrospectively; and the laboratory tension test was limited to the SR most commonly used in our centre. Yet, we performed a biomechanical test on a set of widely used SRs and found their points of failure. The innovative design and superior biomechanical capabilities of the Tigertriever proved successful in the phantom study. Regardless of operator experience, SR mesh can be manually expanded and contracted to conform adequately to the diameter of the affected vessel wall, which decreases the risk of possible complications related to intracranial vessel injury. Furthermore, our experience with the Tigertriever suggests it is positively capable of passing through extremely tortuous, atherosclerotic vessels (a group of patients with the highest risk of SR detachment), but these findings require further investigation. We would like to emphasize that our proposed management of detached stent-retriever removal is optimal for any operator regardless of their skill.

Since the pRESET device detached at the stent–wire interface in vivo, and this was not consistently reproduced in the phantom study, the complication of inadvertent detachment still needs further investigation. We recommend that SR manufacturers should include a tension test in their regular evaluations and inform interventional radiologists how to recognize early signs of possible disconnection.

## 5. Conclusions

Excessive vascular and stent-retriever tension and elongation may precede device disconnection. The point of stent-retriever disconnection is localized mostly at the proximal part of the device. Conservative management of unexpected detachment of a stent retriever during MT of large vessel occlusion may be acceptable in order to avoid further periprocedural complications after failed SR removal attempts. The Tigertriever may be a useful tool for detached SR removal as the phantom study and laboratory test suggests.

## Figures and Tables

**Figure 1 life-11-00658-f001:**
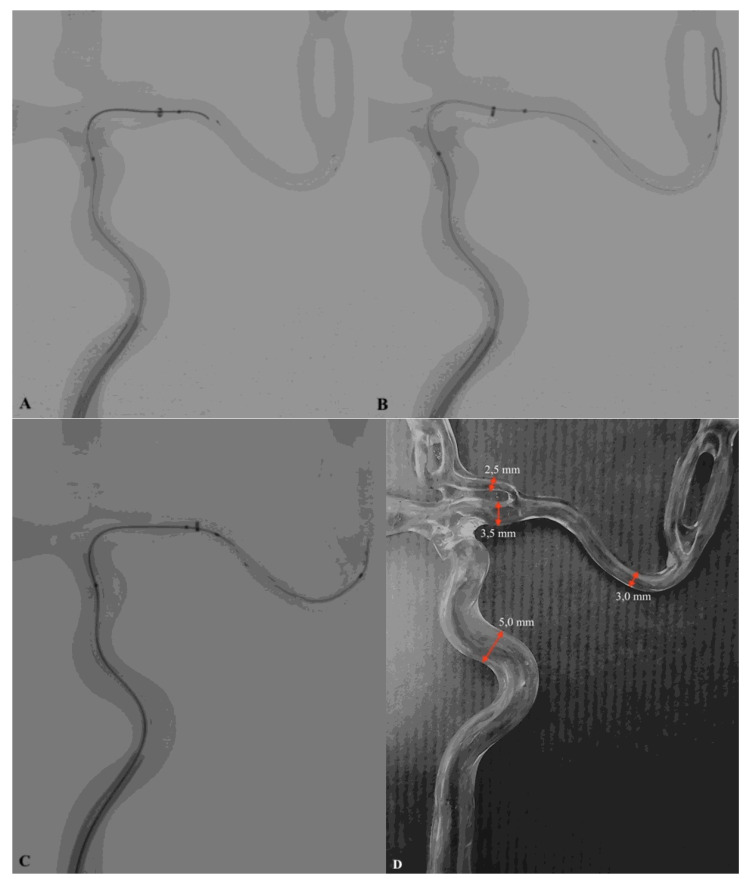
(**A**) Stent retriever pRESET detached in the M1 segment of phantom middle cerebral artery and position of microwire with ‘J’ shape tip, before passing through SR from the vessel. (**B**) Appropriate position of the microwire after crossing the detached SR pRESET lumen. (**C**) Recommended Tigertriever position before starting stent mesh opening and retraction process. (**D**) Vascular phantom model with vessel diameter markings (red arrows). Technical note.

**Figure 2 life-11-00658-f002:**
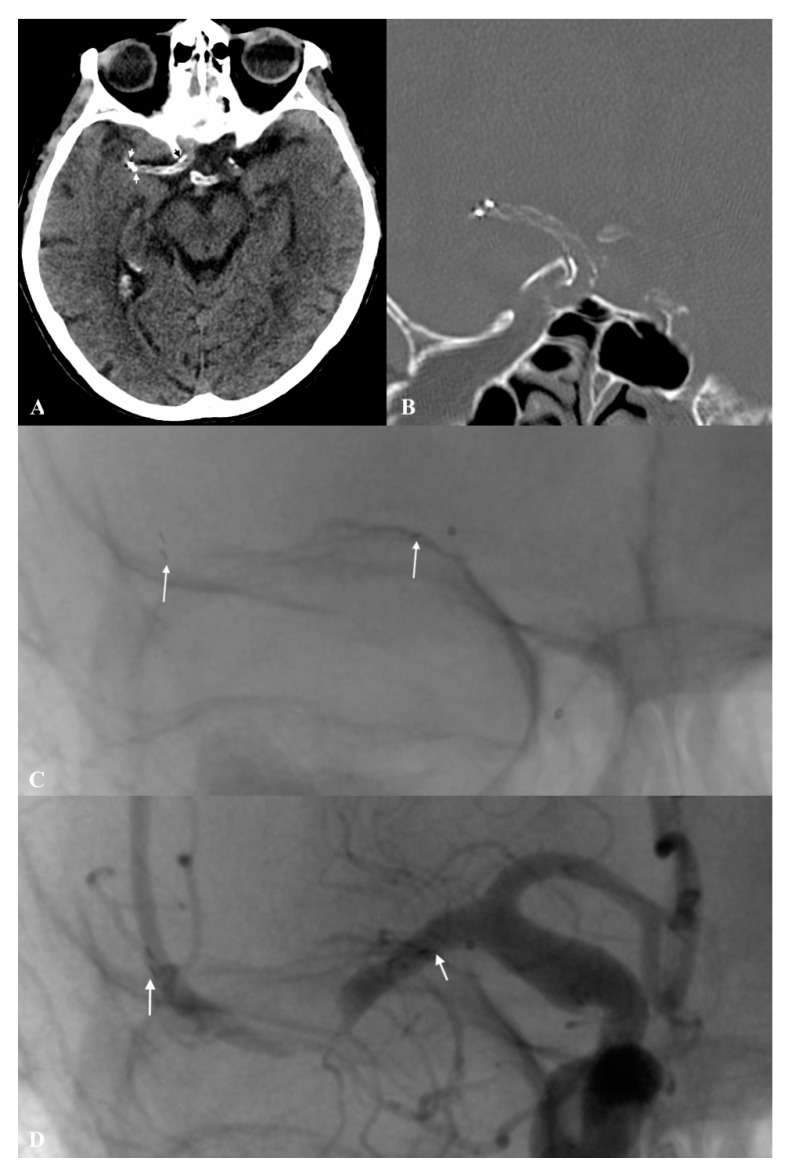
(**A**,**B**) Postprocedural NECT (axial and frontal projections) shows detached stent retriever pRESET 4.0 × 20 mm at the right middle cerebral artery, distal markers (white arrows) and proximal marker (black arrow). (**C**) Stent retriever (pRESET 4.0 × 20 mm) deployed in the M1/2 segments of right middle cerebral artery during first attempt of mechanical thrombectomy with the proximal landing zone in right middle cerebral artery ostium (white arrows). (**D**) Control angiography after pRESET 4.0 × 20 mm detachment (point of detached SR pusher wire and SR distal markers shown with white arrows).

**Figure 3 life-11-00658-f003:**
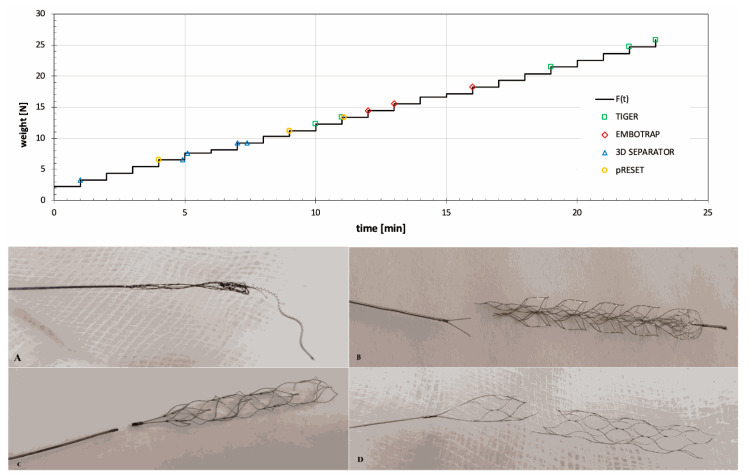
Graph showing the ratio of the SR’s tension force to the load time (the symbols indicate rupture of the tested SRs). The most common rupture types of evaluated stent retrievers during the tension load test: (**A**) Tigertriever, (**B**) Embotrap, (**C**) 3D Separator and (**D**) pRESET.

**Table 1 life-11-00658-t001:** Study data.

Phantom Study							
Interventional Radiologist (No.)	Successful Rescue Procedure Rate (No.)	Average Procedure Time (s)	Procedure Complications				
			pRESET Migration (n)	Breaking of pRESET and/or Tigertriever Structure (n)		Stent Retriever’s Entrapment (n)	Inadvertent Detachment of Tigertriever (n)
1	5/5	165	0	0		0	0
2	5/5	173	0	0		0	0
3	5/5	183	0	0		0	0
**LABORATORY TENSION LOADING TEST**							
**Tested stent retriever**	Tigertriever (RapidMedical)	pRESET (Phenox)	3D Separator (Penumbra)		Embotrap (Johnson&Johnson)		
**Number of tested SR (n)**	5	5	5		3		
**Average tension load causing SR damage (g)**	1992	1093	733		1639		
**Average tension causing SR damage (N)**	19.5	10.7	7.2		16.1		
**Duration of the breaking load (n):**							
when the load was applied	5	4	3		3		
5 s after the load was applied	0	1	1		0		
23 s after the load was applied	0	0	0		0		
55 s after the load was applied	0	0	1		0		
**Type of SR damage (n):**							
deformation of the SR structure, without rupture	3	0	0		0		
SR rupture at 1/3 proximal length	0	2	0		2		
SR rupture at the point of connection with pusher wire (near the proximal stent marker)	0	3	5		0		
pusher wire rupture	2	0	0		0		
SR rupture at the 1/3 distal length	0	0	0		1		

## Data Availability

Not applicable.

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
