# Peer review of "Inadvertent Detachment of Stent Retrievers during Mechanical Thrombectomy—A Clinical and Biomechanical Perspective"

_life, 2021, doi:10.3390/life11070658_

Round 1
Reviewer 1 Report
I want to congratulate the authors for a well written article . Please address the following concerns
- Are there any studies done to to assess the utility of the stent retriever comparing various devices ? If so can you please provide references.
- what would be the clinical significance of your study ? Can you address other variabilities that can impact the superiority of Tigertriever when using in patients?
- Do the experience and training of the interventional radiologist can interfere the outcome?
- Has the laboratory traction load test been used with the stent retriever and how accurate they are? Please include reference for it
- Explain more about the conservative management of inadvertent stent -retriever detachment? can you please provide any article which compared conservative Vs interventional management?
Reviewer 2 Report
The overall study is well conceived within the limits reporting such a rare event (and the consequent small numbers of patients examined).
The manuscript would benefit from additional English proof-reading; in particular, there are a number of times the authors have use "i.e." (a latin acronym meaning "that is") where I'm fairly sure they meant to use "e.g." (meaning "for example").
The authors do not state whether either patient had any atherosclerotic disease or calcification in their arteries in addition to the paroxysmal AF that is mentioned. It would be worthwhile mentioning these factors (or their absence) in the patient descriptions.
Author Response
Please see the attachment.

This manuscript is a resubmission of an earlier submission. The following is a list of the peer review reports and author responses from that submission.